# Lutein/Zeaxanthin Isomers and Quercetagetin Combination Safeguards the Retina from Photo-Oxidative Damage by Modulating Neuroplasticity Markers and the Nrf2 Pathway

**DOI:** 10.3390/ph16111543

**Published:** 2023-11-01

**Authors:** Emre Sahin, Cemal Orhan, Nurhan Sahin, Muralidhara Padigaru, Abhijeet Morde, Mohan Lal, Nanasaheb Dhavan, Fusun Erten, Ahmet Alp Bilgic, Ibrahim Hanifi Ozercan, Kazim Sahin

**Affiliations:** 1Department of Animal Nutrition, Faculty of Veterinary Medicine, Bingol University, Bingol 12000, Turkey; esahin@bingol.edu.tr; 2Department of Animal Nutrition, Faculty of Veterinary Medicine, Firat University, Elazig 23119, Turkey; corhan@firat.edu.tr (C.O.); nsahin@firat.edu.tr (N.S.); 3OmniActive Health Technologies Co., Ltd., Mumbai 400013, India; m.padigaru@omniactives.com (M.P.); a.morde@omniactives.com (A.M.); m.lal@omniactives.com (M.L.); n.dhavan@omniactives.com (N.D.); 4Department of Veterinary Medicine, Pertek Sakine Genc Vocational School, Munzur University, Tunceli 62500, Turkey; fusunerten@munzur.edu.tr; 5Department of Ophtalmology, Ankara Dışkapı Yıldırım Beyazıt Training and Research Hospital, University of Health Sciences, Ankara 06110, Turkey; ahmetalp55@gmail.com; 6Department of Pathology, Faculty of Medicine, Firat University, Elazig 23119, Turkey; ozercanih@firat.edu.tr

**Keywords:** LED light, retinal damage, phytochemicals, plasticity markers, NF-κB, Nrf2

## Abstract

Exposure to light-emitting diode (LED) light is a primary cause of retinal damage, resulting in vision loss. Several plant-derived substances, such as lutein and quercetagetin (QCG), show promise in supporting eye health. In this study, the impact of lutein/zeaxanthin (L/Z, Lutemax 2020) and QCG were evaluated individually and together in a rat model of LED-induced retinal damage. A total of 63 Wistar rats were allocated into nine groups (*n* = 7). For 28 days, the rats received L/Z (10 or 20 mg/kg BW), quercetin (QC, 20 mg/kg BW), QCG (10 or 20 mg/kg BW), or a mixture of different lutein and QCG dosages, after which they were exposed to LED light for 48 h. LED exposure led to a spike in serum malondialdehyde (MDA) and inflammatory cytokines, as well as an increase in retinal NF-κB, ICAM, GFAP, and MCP-1 levels (*p* < 0.0001 for all). It also reduced serum antioxidant enzyme activities and retinal Nrf2, HO-1, GAP43, NCAM, and outer nuclear layer (ONL) thickness (*p* < 0.0001 for all). However, administering L/Z and QCG, particularly a 1:1 combination of L/Z and QCG at 20 mg/kg, effectively reversed these changes. The treatment suppressed NF-κB, ICAM, GFAP, and MCP-1 while enhancing Nrf2, HO-1, GAP43, and NCAM and preventing ONL thickness reduction in LED-induced retinal damage rats. In conclusion, while LED light exposure caused retinal damage, treatment with L/Z, QC, and QCG, particularly a combined L/Z and QCG regimen, exhibited protective effects on the retina. This is possibly due to the modulation of neuroplasticity markers and nuclear transcription factors in the rats’ retinal cells.

## 1. Introduction

Over recent years, LED technology has become increasingly prevalent owing to its superior energy efficiency and the ability to produce high light levels while using less power. Despite these advantages, it is widely recognized that LED lights can trigger phototoxicity in the retina through oxidative stress. This stress predominantly affects photoreceptors and the retinal pigment epithelium (RPE), leading to retinal damage [1]. High-intensity ultraviolet (UV) and blue light exposure can damage the outer segments of photoreceptors, reduce the expression of antioxidative proteins, and increase levels of inflammatory proteins within the retina [2,3].

Earlier studies have demonstrated that carotenoids, including lutein (L), zeaxanthin [4], and β-cryptoxanthin, can help prevent or mitigate retinal degeneration [3]. Among these carotenoids, lutein is the primary macular pigment in the retina. It can protect retinal photoreceptors from blue light damage by boosting macular pigment density [5] and enhancing superoxide dismutase (SOD), glutathione peroxidase (GPX), and catalase (CAT) activities while reducing reactive oxygen species (ROS) production, interleukin-1β (IL-1β), IL-6, tumor necrosis factor-α (TNF-α) [6], and monocyte chemoattractant protein 1 (MCP-1) levels. Lutein can diminish oxidative stress-induced inflammation by promoting nuclear activation of the nuclear factor erythroid 2-related factor 2 (Nrf2) [7]. Owing to its anti-inflammatory properties, lutein may suppress glial fibrillary acid protein (GFAP) expression in Müller glial cells and decrease Müller cell gliosis in the retina [8]. Therefore, consuming lutein in the diet can inhibit oxidative damage in photoreceptors and may reduce the risk of age-related macular degeneration [9]. Quercetin (QC, 3,3′,4′,5,7-pentahydroxyflavanone), a potent antioxidant, is a plant-derived polyphenolic compound commonly found in various fruits and vegetables [10]. Cheng et al. found that QC inhibits monocyte adhesion and attenuates intercellular adhesion molecule-1 (ICAM) expression by suppressing TNF-α and nuclear factor kappa B (NF-κB) in RPE cells [11]. Moreover, QC can increase the expression of ROS-catalyzing phase II proteins such as heme oxygenase-1 (HO-1) by inducing the antioxidant transcription factor Nrf2 in H_2_O_2_-induced oxidative damage in RPE cells [12]. Therefore, QC can alleviate light-induced retinal damage due to its ability to stimulate antioxidant enzymes [13] and suppress inflammatory cytokines [14]. Quercetagetin (QCG, 3,3′,4′,5,6,7-hydroxyflavone) has a structure very similar to QC. However, QCG may demonstrate stronger radical scavenging and antioxidant activity than QC [15]. Additionally, an in vitro study indicated that QCG’s anti-inflammatory activity might be greater than QC [16].

Numerous studies suggest that the combined use of lutein [4,17] or QC with other antioxidants can enhance their retinal protective effects. This combined administration may provide more potent protection against oxidative damage to retinal cells than when administered separately. Though previous research has demonstrated the efficacy of QCG in combating oxidative damage when administered individually, to our knowledge, there is limited investigation into the possible protective impact of combining L/Z and QCG on damage to the retina. Therefore, the objective of this study is to explore the potential protective role of L/Z and QCG combination on retinal damage, neuronal plasticity, and regenerative potential in a rat model exposed to intense LED light. Simultaneously, this study will also examine alterations in neuroplasticity markers, such as neural cell adhesion molecule (NCAM), growth-associated protein-43 (GAP43), and GFAP, as well as nuclear transcription factors. This research seeks to address the current knowledge gap regarding the potential synergistic benefits of the L/Z and QCG combination in shielding the retina from oxidative damage from LED light exposure.

## 2. Results

### 2.1. Serum Malondialdehyde (MDA) and Antioxidant Enzyme Levels

Compared to the control group, the LED group showed a rise in serum MDA levels (Figure 1A, *p* < 0.0001) and a decline in serum antioxidant enzyme activities (SOD, CAT, and GPX) (Figure 1B–D *p* < 0.0001). L/Z, QC, QCG, and their combinations effectively counteracted the rise in retinal MDA levels caused by LED exposure (*p* < 0.0001). The L + QCG (1:1) combination was most effective in lowering retinal MDA levels (*p* < 0.0001). Furthermore, the LED + L/Z + QCG (1:1) group presented elevated serum SOD [(*p* < 0.05 for the LED + L/Z + QCG (1:0.5) group and *p* < 0.0001 for other groups), CAT (*p* < 0.01 for the LED + L/Z + QCG (1:0.5) group and *p* < 0.0001 for other groups)], and GPX activities when compared to retinal damage groups [(*p* < 0.001 for the LED + L/Z + QCG (1:0.5) group and *p* < 0.0001 for other groups)]. These findings showed that L + QCG (1:1) was more effective in improving serum antioxidant enzymes compared to L + QCG (1:0.5) and L + QCG (0.5:0.5).

### 2.2. Serum Inflammatory Cytokine Levels

Figure 2 demonstrates that rats exposed to LED light for 48 h exhibited higher inflammatory cytokine levels (IL-1β, IL-6, and TNF-α) compared to healthy control rats (*p* < 0.0001, for all). While the groups treated with either L/Z or QCG (20 mg/kg) displayed similar inflammatory cytokine levels, their various dose combinations significantly lowered these levels. The L/Z and QCG combination (at a 1:1 ratio) more effectively reduced serum IL-6 levels in LED-exposed rats than in other formulations (Figure 2B, *p* < 0.0001). There was no significant difference in IL1-β and TNF-α levels between the LED + L/Z + QCG (1:1) and LED + L/Z + QCG (1:0.5) groups. Furthermore, the suppressive activity on inflammatory cytokines was higher in rats given high-dose L/Z and QCG (at a 1:1 ratio) compared to those receiving low-dose L/Z and QCG (at a 0.5:0.5 ratio) (*p* < 0.001 for TNF-α and *p* < 0.0001 for IL-1β and IL-6).

### 2.3. Retinal Protein Levels

We found that the rats exposed to LED for 48 h had higher levels of B-cell lymphoma-2 protein (Bcl-2) and lower levels of Bcl-2-associated X protein (Bax) and cysteine–aspartic acid protease-3 (Caspase-3) in retinal tissue compared to the healthy control rats (Figure 3A–C, *p* < 0.0001). The combined administration of L/Z and QCG (at a 1:1 ratio) more effectively improved retinal Bax (*p* < 0.01 for the LED + L/Z + QCG (1:0.5) group and *p* < 0.0001 for others) and Caspase-3 activity than other formulations (*p* < 0.05 for the LED + L/Z + QCG (1:0.5) group and *p* < 0.0001 for others). Conversely, the combination of L/Z and QCG at a 1:1 ratio significantly reduced Bcl-2 activity compared to other formulations [(*p* < 0.001 for the LED + L/Z + QCG (1:0.5) group and *p* < 0.0001 for other groups)].

We measured the transcription factors Nrf2, HO-1, and NF-κB levels to assess the potential antioxidant and anti-inflammatory activity of the L/Z and QCG combination in the retinal tissue (Figure 4A–C). As expected, we found that retinal Nrf2 and Nrf2-related protein HO-1 were significantly downregulated in rats exposed to LED compared to control rats (Figure 4A,B, *p* < 0.0001 for both). In contrast, retinal NF-κB levels, which mediate inflammation and oxidative stress, were increased by LED-induced retinal damage in rats. Although retinal Nrf2 levels did not statistically different between the QC, QCG2, L + QCG (1:0.5), or L + QCG (0.5:0.5) administrated groups, the LED + L/Z + QCG (1:1) group had higher Nrf2 levels than other LED-exposed groups (*p* < 0.001 for the LED + L/Z + QCG (1:0.5) group and *p* < 0.0001 for others). Similarly, retinal OH-1 levels were relatively higher in the LED + L/Z + QCG (1:1) group than in the other LED groups (*p* < 0.0001). Although there was no notable change in NF-κB levels between the LED + L/Z + QCG (1:0.5) and LED + L/Z + QCG (1:1) groups, we found that these groups had higher NF-κB levels than the other LED groups (*p* < 0.01 for LED + L/Z + QCG (1:0.5) vs. LED + L/Z + QCG (0.5:0.5); *p* < 0.0001 for LED + L/Z + QCG (1:0.5) vs. other groups; *p* < 0.0001 for LED + L/Z + QCG (1:1) vs. other groups). Concurrently with the elevation in serum antioxidant capacity, the L + QCG (1:1) formulation exhibited a distinct enhancement in retinal Nrf2 and HO-1 levels in comparison to both the L + QCG (1:0.5) and L + QCG (0.5:0.5) formulations.

Intense LED light exposure led to the upregulation of retinal ICAM, GFAP, and MCP-1 levels, while GAP43 and NCAM levels were found to be downregulated (Figure 5A–E, *p* < 0.0001). A notable decrease in ICAM and MCP-1 protein levels was observed in the retinas of rats given a combination of L/Z and QCG (in a 1:1 ratio) at a dose of 20 mg/kg, compared to rats receiving other formulations (*p* < 0.0001 for all). Additionally, there was no significant difference in GFAP levels between the LED + L/Z + QCG (1:0.5) and LED + L/Z + QCG (1:1) groups (*p* > 0.05), and these two groups exhibited the lowest GFAP levels compared to other LED-exposed rats (*p* < 0.0001 for all). We found that rats supplemented with the combined L/Z and QCG (formulated in a 1:1 ratio) had significantly elevated retinal GAP43 and NCAM levels compared to the retinal damage groups, including the LED + L/Z + QCG (1:0.5) and LED + L/Z + QCG (0.5:0.5) groups. These findings emphasized that the regulatory effect of L + QCG (1:1) on retinal neuroplasticity may be more effective than other formulations.

### 2.4. Retinal Histopathology

Figure 6 demonstrates that the histological structure of the retina was normal in control rats. In contrast, intense LED light exposure led to significant severe edema (+++) in the ganglion layer of the retina and thickening of the outer plexiform layer in rats. The administration of L/Z and QCG, particularly when combined, effectively prevented these abnormalities compared to non-supplemented retinal damage rats. No notable increase in outer nuclear layer (ONL) thickness was observed when L/Z, QC, or QCG were administered individually (*p* > 0.05). We found that L/Z and QCG, formulated at a ratio of 1:1 or 1:0.5 and administered at a dose of 20 mg/kg, prevented the decrease in ONL thickness compared to their individual administration (*p* < 0.01). The LED + L/Z + QCG (1:1) and LED + L/Z + QCG (1:0.5) groups exhibited very mild edema (+), which appeared to be nearly normal, in the ganglion layer of the retina.

## 3. Discussion

Our findings confirmed that subjecting rats to intense LED exposure for 48 h was a potent inducer of retinal damage, consistent with earlier research [3,18]. LED light can lead to photo-oxidative stress, causing a loss of photoreceptors and a significant decrease in ONL thickness [6,19]. Orhan et al. found increased MDA, an indicator of lipid peroxidation, levels in both the retina and blood serum after LED exposure [18]. Similar to our results, Wang et al. discovered that light-induced retinal damage suppressed the activity of serum antioxidant enzymes SOD and GPX [20]. Moreover, we found that serum inflammatory cytokines (IL-1β, IL-6, and TNF-α) were increased due to intense LED light exposure. These findings are consistent with Yang et al. suggesting that the level of inflammatory cytokines in serum is considered a sensitive indicator of the degree of retinal inflammation [6]. Depending on its ROS scavenge activity, lutein can enhance anti-inflammatory responses by inhibiting NF-κB-related cytokines and inflammatory enzymes like cyclooxygenase-2 and inducible nitric oxide synthase [7]. Similarly, Chen et al. demonstrated that lutein intervention at different doses (12.5, 25, and 50 mg/kg) could mitigate oxidative damage and decrease inflammatory cytokine production in rat serum [21]. Our prior research showed that oral lutein/zeaxanthin isomers reduced retinal MDA concentration while increasing SOD, CAT, and GPX activity in diabetic [22] and photo-oxidative retinal damage rats [4]. Kumar et al. found that QC (25 and 50 mg/kg) effectively inhibited retinal NF-κB-related inflammatory cytokine release and improved antioxidant enzyme function in streptozotocin-induced diabetic rats [13]. These studies corroborate our findings, demonstrating that lutein and QC reduce oxidative stress and suppress inflammatory cytokine activities. An in vitro study by Kang et al. indicated that the inflammatory chemokine suppressive activity of QCG could be higher than QC in human keratinocytes [16]. Bulut and Deniz Yilmaz reported that QCG showed stronger antioxidant activity than QC, butylated hydroxyanisole (BHA), and Trolox in DPPH and ABTS radical scavenging assays [15]. The current study supported these earlier in vitro findings and revealed for the first time that 20 mg/kg of QCG was more effective than 20 mg/kg of QC in inhibiting inflammatory cytokines in rat serum after LED exposure. Therefore, it was clarified that the antioxidant effectiveness of the QCG is higher than QC in rats. Additionally, here we reported that the combination of L/Z and QCG, particularly at a 1:1 ratio (20 mg/kg), synergistically reduced systemic oxidative stress and inflammation in rats. However, due to its specific experimental design, the current study is limited, as it did not allow for an assessment of the relationship between systemic oxidative stress and retinal damage in rats.

Increased ectopic expression of Bcl-2 against retinal damage protects photoreceptor cells from oxidative toxicity in transgenic retinal degeneration slow mice [23]. Godley et al. reported that Bcl-2 overexpression protects RPE cells from H_2_O_2_-induced oxidative DNA damage and promotes cell survival [24]. These studies suggest that Bcl-2 and Bax may be differently regulated to protect the retina against retinal damage inducers. In line with previous research, we demonstrated that LED exposure increased retinal Bcl-2 levels and decreased retinal Bax and Caspase-3 levels in rats [3,18]. This is likely because the increased Bcl-2 level may antagonize proapoptotic Bax [25] and Caspase-3 activation [26]. While there is limited research on the impact of L/Z and QCG on Bax and Bcl-2 levels in rats exposed to LED light, Zhang et al. demonstrated that lutein could support photoreceptor survival in retinitis pigmentosa by enhancing the expression of rhodopsin and opsin in cells [8].

Furthermore, Sahin et al. found that lutein/zeaxanthin isomers increased rhodopsin mRNA levels in rats exposed to LED light for 24 h [4]. It has also been reported that oral administration of β-Cryptoxanthin, an essential dietary carotenoid, for 28 days resulted in reduced retinal Bcl-2 levels and elevated Bax and Caspase-3 levels in rats exposed to intense LED light for 48 h [3]. An in vitro study by Zhu et al. demonstrated that QC could decrease the cigarette smoke extracts induced apoptosis rate in RPE cells [27]. Previously, it was suggested that 50 μM/L QCG had an inhibitory effect (80%) on DNA damage in Vero cells [28]. No prior research is currently available that has examined the impact of QCG on retinal apoptosis markers. The current study demonstrated that administering QCG, especially when combined with L/Z (1:1 ratio), might prevent retinal damage in rats by modulating the Bcl-2/Bax/Caspase-3 interaction.

As anticipated, intense LED exposure suppressed retinal Nrf2 and HO-1 protein activity while stimulating NF-κB production. These findings align with previous studies on LED-induced retinal damage models in rodents, which involved exposure for 24 or 48 h [3,4,29]. An in vitro study by Shivarudrappa and Ponesakki reported that in RPE cells, lutein protected the high glucose-induced down-regulation of a redox-sensitive transcription factor, Nrf2, as well as antioxidant enzymes, including SOD2, HO-1, and CAT [30]. Tuzcu et al.’s study in a diet-induced obesity rat model demonstrated that supplementation with lutein/zeaxanthin isomers could modulate retinal NF-κB and Nrf2/HO-1 signaling pathways, thereby alleviating retinal oxidative stress [22]. In a separate study, Gunal et al. showed that lutein/zeaxanthin isomer supplementation at 20 mg/kg reduced NF-κB levels and increased Nrf2 in the brain of a traumatic brain injury mouse model [31]. These earlier studies corroborated our findings, which demonstrate that lutein can activate Nrf2 and increase the expression of antioxidant enzymes, including HO-1, in the retinas of rats exposed to LED light. Likewise, QC and QCG may support antioxidant capacity by upregulating Kelch-like ECH-associated protein 1 (Keap1)/Nrf2/antioxidant responsive element (ARE) pathway and increasing antioxidant enzymes in various tissues, including the retina [27] and liver [32]. Zhu et al. showed that QC could modulate the Keap1/Nrf2/ARE pathway and inflammation induced by cigarette smoke extracts in RPE cells [27]. Additionally, a study by Kumar et al. revealed that QC (25 and 50 mg/kg) decreased retinal NF-κB activity in diabetic rats [13]. Although no studies directly compare the effects of QCG on retinal Nrf2, HO-1, and NF-κB, Wu et al. demonstrated that dietary QCG supplementation (3.2, 4.8, or 6.4 mg/kg diet) enhanced the Nrf2/HO-1 signaling pathway in chicken livers [32]. Furthermore, an in vitro study by Baek et al. found that QCG dose dependently (5, 10, and 20 μM) reduced UVB-induced NF-κB transactivation in epidermal JB6 P+ cells [33]. In alignment with previous research, our results indicate that both QC and QCG boost Nrf2 activity. Furthermore, our findings showed that QCG has a more potent capacity to stimulate Nrf2 than L/Z. Consequently, combining L/Z and QCG in varying proportions, particularly 1:1, increased antioxidant capacity better than administration alone, likely owing to their cumulative antioxidant activity. Also, the elevated antioxidant enzyme, SOD, CAT, and GPX levels supported that L/Z and QCG administration may exhibit retinal tissue protection against LED light-induced retinal damage by regulating transcription factor Nrf2.

An earlier in vitro study indicated that damage caused by LED light might raise ICAM levels in 661W photoreceptor cell lines [34]. Another investigation discovered that prolonged LED-induced photo-oxidative stress could trigger the upregulation of retinal MCP-1 through phospholipid oxidation in mice [35]. The level of GFAP, a neuronal stress marker found in Müller glial cells, has been observed to increase in the retinas of chickens exposed to 2000 lux fluorescent light for 23 days [36]. Consistent with the present study’s findings, previous research has shown a decrease in retinal GAP43 and NCAM activity. Additionally, a positive association has been observed between elevated GFAP expression and retinal degeneration, all attributed to exposure to LED light [3,4,37]. On the other hand, GAP43 aids in photoreceptor survival and delays retinal degeneration, and a deficiency in this protein may potentially impact retinal plasticity in rats [38]. Furthermore, NCAM, involved in sustaining plasticity within the visual pathway, may help defend the retina against oxidative damage by fostering cell survival and maintaining the retina’s physiological processes [37,39]. Previous research has indicated that lutein/zeaxanthin isomers may reduce GFAP activity while enhancing GAP43 and NCAM protein levels in light-induced retinal damage in rats [4]. Additionally, in vitro studies conducted by Cheng et al. [11] and Cheng et al. [40] proposed that QC could decrease the production of ICAM, IL-6, IL-8, and MCP-1 by inhibiting the activation of NF-κB and TNF-α signaling pathways in human retinal epithelial cells. In agreement with these data, we observed that L/Z and QC elevated the tissue-protective GAP43 and NCAM levels while inhibiting the increase in retinal ICAM, GFAP, and MCP-1 proteins. Presently, there is a lack of studies available to evaluate the impact of QCG on ICAM, GFAP, MCP-1, GAP43, and NCAM. The NF-κB suppressing and Nrf2 stimulating properties of QCG could potentially account for its effects on these markers. As a result of the inhibitory impacts of both L/Z and QCG on NF-κB, the combination of L/Z and QCG (in ratios of 1:1 and 1:0.5) is likely to exhibit greater efficacy in counteracting the activity of ICAM, GFAP, and MCP-1 in rat retinas against photo-oxidative stress. On the other hand, the Nrf2-related antioxidant function of L/Z and QCG might contribute to enhancing GAP43 and NCAM-related retinal plasticity. Consequently, we demonstrated that L/Z plus QCG (formulated at 1:1 or 1:0.5 ratios) administration could prevent the reduction of ONL thickness and protect retinal integrity by regulating retinal plasticity markers and antioxidant transcription factors.

## 4. Materials and Methods

### 4.1. Animals and Experimental Procedures

A total of 63 Wistar albino female rats (age: 8 weeks, weight: 180 ± 20 g) were housed in a controlled environment at a temperature of 22 °C with a 12:12 h light–dark cycle and provided with ad libitum rat chow and water. Statistical power (1-β = 85%), effect size (f = 0.59), and total sample size (*N =* 63) were determined by power analysis in the G* Power program (Version 3.1) [41]. All experiments were carried out in compliance with the National Institutes of Health’s Guidelines for the Care and Use of Laboratory Animals and received approval from Bingol University Local Ethics Committee (2022/03-03/01) in accordance with EU directives. This study was reported according to Animal Research: Reporting of In Vivo Experiments (ARRIVAL) guidelines.

### 4.2. Experimental Procedures

After a week-long acclimation period, the rats were randomly divided into nine different treatment groups, each consisting of seven rats, as outlined in Table 1. All the products used were provided by OmniActive Health Technologies Ltd. (Pune, India). The product compositions are detailed in Table 2. These products are natural extracts, standardized with stabilizers and natural antioxidants, and derived from the dried flower of the Marigold plant (*Tagetes erecta*). It has been previously reported that doses higher than the doses used in this study have not been found to cause chronic toxicity in animals or humans and that none of the compounds produces mutagenicity [42,43]. Corn oil (0.5 mL/rat/day) was used as a vehicle to dissolve lutein/zeaxanthin (L/Z, Lutemax 2020, OmniActive Health Technologies Ltd., Pune, India), QC, and QCG. The substances were administered by oral gavage with a metal feeding needle. The control and LED groups were given only corn oil (0.5 mL/rat/day) for 28 days. The gavage process was conducted every alternate day for 28 days. The doses of L/Z and QC were based on previous studies by Gunal et al. [31] and Sefil et al. [44].

After 28 days to evaluate the protective effects of compounds, all groups, excluding the control group, were subjected to intense LED light for 48 h to cause retinal degeneration. The Retinal damage model was established according to Orhan et al. [18]. The animal’s pupils were dilated once with 1% tropicamide before light exposure. Rats were accommodated in isolated transparent light stress boxes equipped with white LED (750 lux) on top of the shelves and 20 cm away from the source. This arrangement ensured consistent inner luminance. Rat chow and water were provided ad libitum. After exposure to LED light, the animals were kept in a dark room until sacrificed. Next, rats were anesthetized (10 mg/kg xylazine and 50 mg/kg ketamine) and euthanized by cervical dislocation. Blood samples of each animal were collected into clot activator vacuum tubes to extract serum. The right eyes of each animal were immediately transferred into a 4% paraformaldehyde solution. The left eyes were gently dissected to remove retinas, and the collected retinas were stored at −80 °C until laboratory analysis.

### 4.3. Biochemical Analyses

Blood samples were centrifuged at 4 °C at 3000× *g* for 10 min, and serum samples were extracted. Serum SOD (MBS266897), CAT (MBS006963), and GPX (MBS701677, MyBioSource, San Diego, CA, USA) activity and inflammatory cytokine [IL-1β (E-UNEL-R0028, IL-6 (E-EL-R0015), and TNF-α (E-EL-R2856; Elabscience Biotechnology Inc., Wuhan, China) concentrations were determined with the rat-specific kits according to the manufacturers’ guidelines in a microplate reader (Elx-800, Bio-Tek Instruments Inc., Winooski, VT, USA).

Serum MDA levels were measured by High-performance Liquid Chromatography (HPLC, Shimadzu, Kyoto, Japan) according to the method described by Orhan et al. [3]. Briefly, samples were homogenized with 0.5 M HClO_4_ and 2[6]-di-tert-butyl-p-cresol, and supernatants were obtained after centrifugation. Supernatants were loaded (20 μL) into the HPLC system [UV–vis SPD-10 AVP detector, CTO-10 AS VP column, 30 mM KH_2_PO_4_: CH_3_OH (82.5:17.5, *v*/*v*, pH 3.6) mobile phase at a flow rate of 1 mL/min], and chromatograms were scanned at 250 nm.

### 4.4. Western Blot Analysis

Bcl-2, Bax, Caspase-3, Nrf2, HO-1, NF-κB p65, ICAM, GAP43, GFAP, NCAM, and MCP-1 levels were detected by Western blotting, as defined previously [3]. Briefly, homogenized retina tissues were mixed with 2xLaemmli buffer and boiled for 5 min before electrophoresis. Retinal protein samples were run in 12% sodium dodecyl sulfate (SDS)–polyacrylamide gel (PAGE) electrophoresis. Next, proteins were transferred to the nitrocellulose membrane (pore size 0.45 μm). Nitrocellulose membranes blocked with bovine serum albumin (5%) and then incubated with diluted (1:1000) primary antibodies [Bcl-2 (ab196495), Bax (ab32503), Caspase-3 (ab13585), HO-1 (ab13243), NF-κB p65 (ab19870), GAP43 (ab16053), NCAM (ab95153), MCP-1 (ab25124), and β-actin (ab8227; Abcam, Cambridge, UK). Nrf2 (#33649) and GFAP (#80788); Cell Signaling Technology, Danvers, MA, USA. ICAM (sc-8439); Santa Cruz Biotechnology, Heidelberg, Germany)] for overnight. The membranes were washed and moved into the appropriate HRP-conjugated secondary antibody (1:2500, ab205719, ab6721; Abcam, Cambridge, UK) solution. Protein loading was ensured with β-actin antibody. The interaction between primary and secondary antibodies was visualized by the diaminobenzidine substrate method. Visualized bands were densitometrically analyzed with Image J software (Version 1.4.3.67, National Institute of Health, Bethesda, MD, USA).

### 4.5. Histopathological Analysis

Retina samples were fixed in 4% paraformaldehyde for histopathological evaluation and embedded in paraffin. Paraffin blocks were sectioned by 5 μm slices and stained according to routine hematoxylin and eosin (H&E) procedures. The sections were obtained along the vertical meridian, enabling the comparison of all retinal regions in both the superior and inferior hemispheres. Histological analyses were conducted by examining the mid-superior section of the tissue using a light microscope. The ONL thickness was measured by morphometry along the vertical meridian in stained sections of the retina by an independent blinded pathologist, as defined earlier [45]. Retinal edema has been classified based on severity as follows: (+) for mild edema, (++) for moderate edema, and (+++) for severe edema.

### 4.6. Statistical Analyses

All data are represented as mean and standard deviation. Shapiro–Wilk test was used for controlling the conformity to the normal distribution of data, whereas Levene’s test was for homogeneity of the variances. For parametric data, multiple comparisons were performed by one-way ANOVA and Bonferroni’s multiple comparisons test. *p* < 0.05 level was considered a criterion of statistical significance. SPSS statistical package program (version 22.0, IBM, SPSS, Armonk, NY, USA) was used for statistical analyses.

## 5. Conclusions

In conclusion, exposure to LED light led to an increase in oxidative stress and a decrease in antioxidant enzyme activities. However, the combination of lutein, QC, and QCG effectively mitigated this increase in retinal oxidative stress. The combination of lutein and QCG in a 1:1 ratio was the most successful in reducing retinal oxidative stress. This combination also amplified retinal antioxidant activity by raising Nrf2 and HO-1 levels while reducing NF-κB levels. This data suggests that lutein and QCG may be potentially beneficial as a treatment approach for LED-induced retinal damage. However, more research is required to ascertain the ideal dosage and duration of supplementation in a clinical context.

## Figures and Tables

**Figure 1 pharmaceuticals-16-01543-f001:**
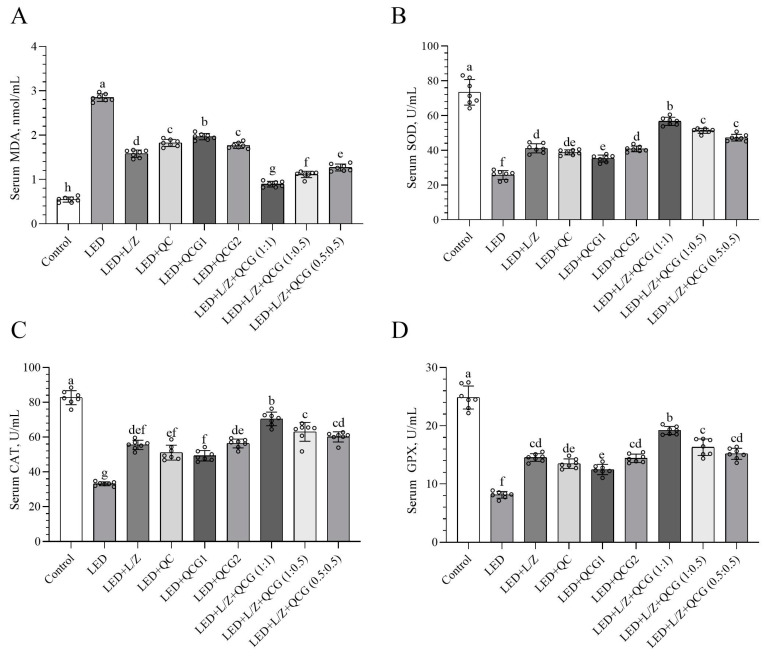
Effect of lutein/zeaxanthin (L/Z) and quercetagetin (QCG) on serum malondialdehyde (MDA, Panel (**A**)); superoxide dismutase (SOD, Panel (**B**)); catalase (CAT, Panel (**C**)); and glutathione peroxidase (GPX, Panel (**D**)) levels in LED-induced retinal damage in rats. The depicted bars and error lines represent mean ± standard deviation for groups. Different small letters (a–h) above the bars indicate statistical differences between groups. There is no difference between groups with the same letter, whereas there is a difference between groups with entirely different letters (*n* = 7, ANOVA and Bonferroni’s multiple comparisons test; *p* < 0.05). The pairwise comparison is presented in Appendix A.

**Figure 2 pharmaceuticals-16-01543-f002:**
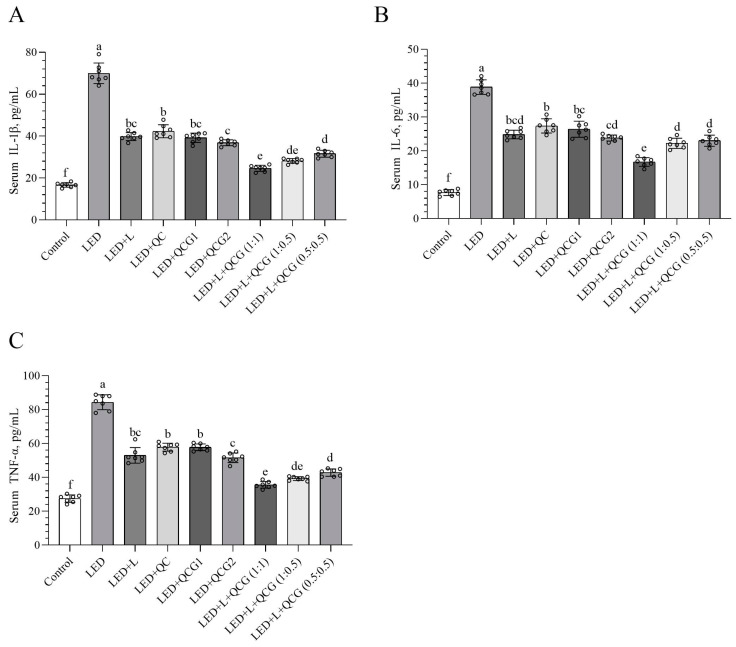
Effect of lutein/zeaxanthin (L/Z) and quercetagetin (QCG) on serum interleukin-1β (IL-1β, Panel (**A**)); IL-6 (Panel (**B**)); and tumor necrosis factor-α (TNF-α, Panel (**C**)) levels in LED-induced retinal damage in rats. The depicted bars and error lines represent mean ± standard deviation for groups. Different small letters (a–f) above the bars indicate statistical differences between groups. There is no difference between groups with the same letter, whereas there is a difference between groups with entirely different letters (*n* = 7, ANOVA and Bonferroni’s multiple comparisons test; *p* < 0.05). The pairwise comparison is presented in Appendix A.

**Figure 3 pharmaceuticals-16-01543-f003:**
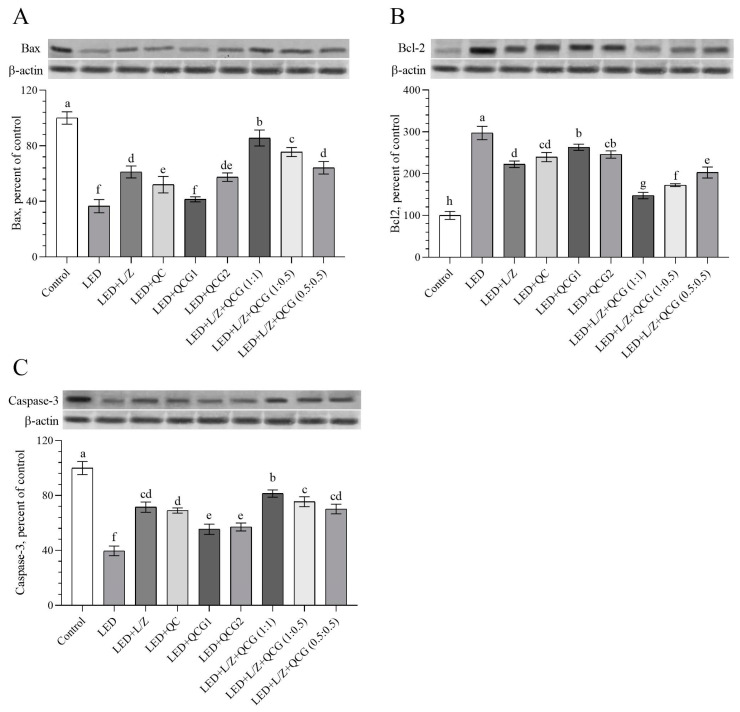
Effect of lutein/zeaxanthin (L/Z) and quercetagetin (QCG) on retinal Bcl-2-associated X protein (Bax, Panel (**A**)); B-cell lymphoma 2 (Bcl-2, Panel (**B**)); and cysteine–aspartic acid protease-3 (Caspase-3, Panel (**C**)) levels in LED-induced retinal damage in rats. β-actin was used to ensure equal protein amounts. The Western blot bands’ intensity is represented according to the percent of the control group. A representative blot is shown in each panel. The depicted bars and error lines represent mean ± standard deviation for groups. Different small letters (a–h) above the bars indicate statistical differences between groups. There is no difference between groups with the same letter, whereas there is a difference between groups with entirely different letters (*n* = 7, ANOVA and Bonferroni’s multiple comparisons test; *p* < 0.05). The pairwise comparison is presented in Appendix A.

**Figure 4 pharmaceuticals-16-01543-f004:**
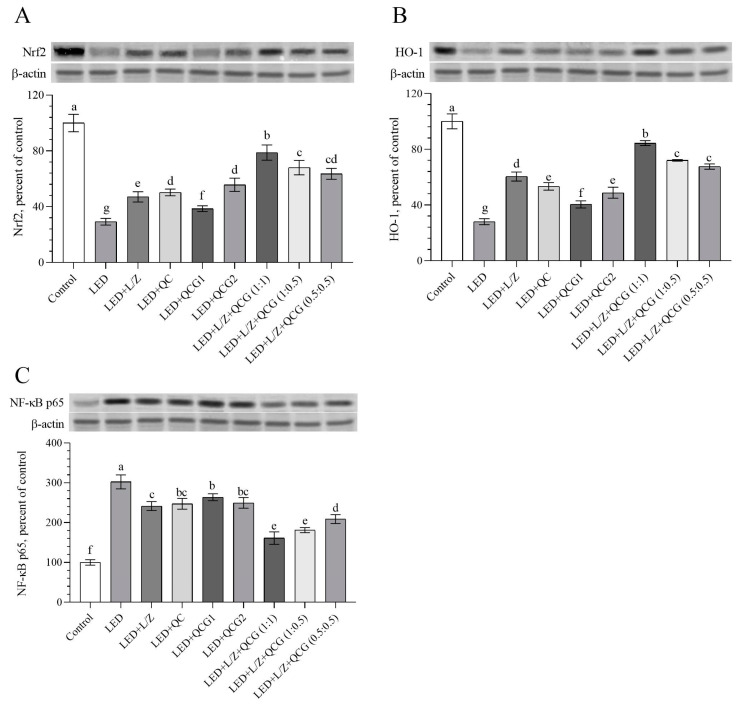
Effect of lutein/zeaxanthin (L/Z) and quercetagetin (QCG) on retinal nuclear factor erythroid 2-related factor 2 (Nrf2, Panel (**A**)); heme oxygenase-1 (HO-1, Panel (**B**)); and nuclear factor kappa B p65 subunit (NF-κB p65, Panel (**C**)) levels in LED-induced retinal damage in rats. β-actin was used to ensure equal protein amounts. The Western blot bands’ intensity is represented according to the percent of the control group. A representative blot is shown in each panel. The depicted bars and error lines represent mean ± standard deviation for groups. Different small letters (a–g) above the bars indicate statistical differences between groups. There is no difference between groups with the same letter, whereas there is a difference between groups with entirely different letters (*n* = 7, ANOVA and Bonferroni’s multiple comparisons test; *p* < 0.05). The pairwise comparison is presented in Appendix A.

**Figure 5 pharmaceuticals-16-01543-f005:**
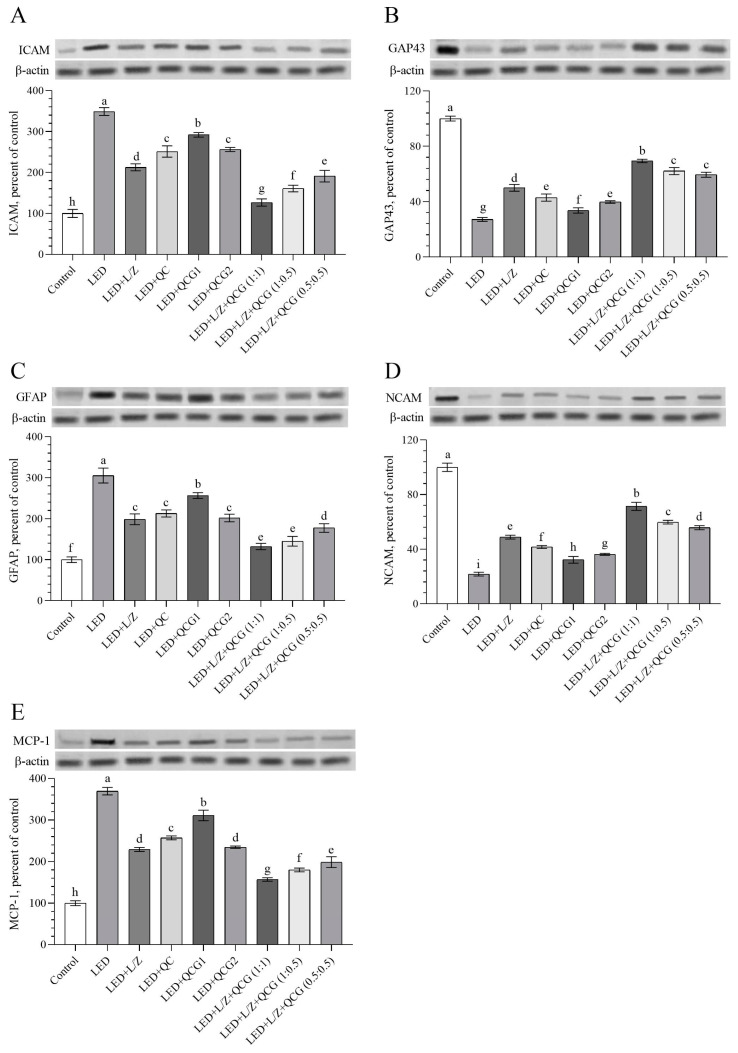
Effect of lutein/zeaxanthin (L/Z) and quercetagetin (QCG) on retinal intercellular adhesion molecule-1 (ICAM, Panel (**A**)); growth-associated protein-43 (GAP43, Panel (**B**)); glial fibrillary acid protein (GFAP, Panel (**C**)); neural cell adhesion molecule (NCAM, Panel (**D**)); and monocyte chemoattractant protein 1 (MCP-1, Panel (**E**)) levels in LED-induced retinal damage in rats. β-actin was used to ensure equal protein amounts. The Western blot bands’ intensity is represented according to the percent of the control group. A representative blot is shown in each panel. The depicted bars and error lines represent mean ± standard deviation for groups. Different small letters (a–i) above the bars indicate statistical differences between groups. There is no difference between groups with the same letter, whereas there is a difference between groups with entirely different letters (*n* = 7, ANOVA and Bonferroni’s multiple comparisons test; *p* < 0.05). The pairwise comparison is presented in Appendix A.

**Figure 6 pharmaceuticals-16-01543-f006:**
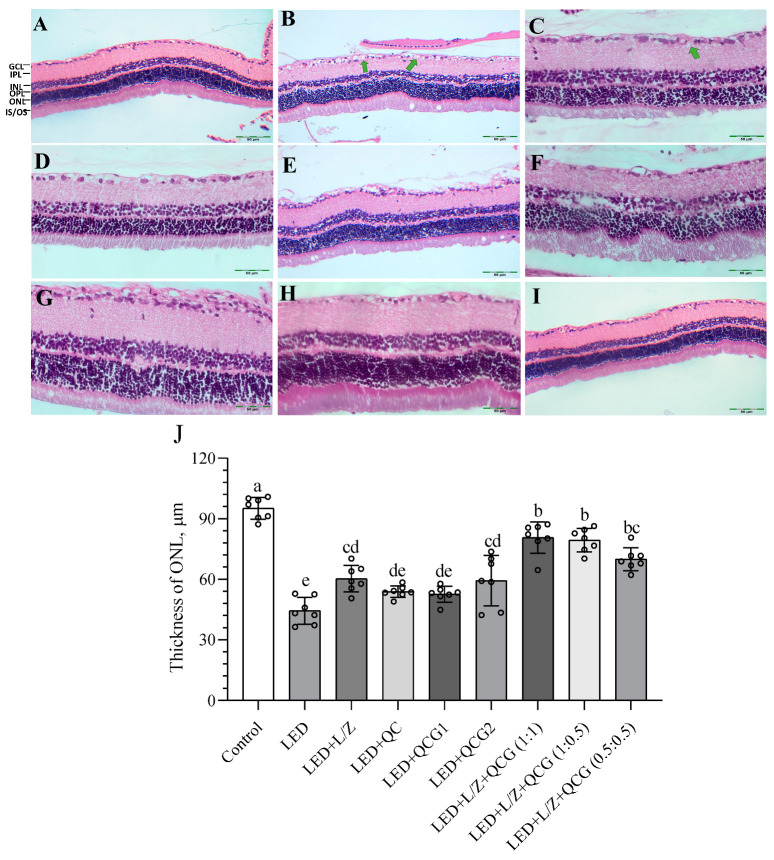
Effect of lutein/zeaxanthin (L/Z) and quercetagetin (QCG) on retinal histopathology [Control, Panel (**A**); LED Panel (**B**); LED + L/Z, Panel (**C**); LED + QC, Panel (**D**); LED + QCG1, Panel (**E**); LED + QCG2, Panel (**F**); LED+ L/Z + QCG (1:1) Panel (**G**); LED+ L/Z + QCG (1:0.5) Panel (**H**); LED+ L/Z + QCG (0.5:0.5), Panel (**I**)] and outer nuclear layer (ONL, Panel (**J**)) thickness in LED-induced retinal damage in rats. Representative images are shown (H&E X200, green arrow: edema). GCL: ganglion cell layer; IPL: inner plexiform layer; INL: inner nuclear layer; OPL: outer plexiform layer; ONL: outer nuclear layer; IS/OS: layer of rods and cones. The depicted bars and error lines represent mean ± standard deviation for groups. Different small letters (a–e) above the bars indicate statistical differences between groups. There is no difference between groups with the same letter, whereas there is a difference between groups with entirely different letters (n = 7, ANOVA and Bonferroni’s multiple comparisons test; *p* < 0.05). The pairwise comparison is presented in Appendix A.

**Table 1 pharmaceuticals-16-01543-t001:** Experimental groups.

Groups	Treatments
Control	Not exposed to intense LED light and administered with 0.5 mL vehicle (corn oil)
LED	Exposed to intense LED light and administered with 0.5 mL of vehicle
LED + L/Z	Exposed to intense LED light and administered with 20 mg/kg BW of L/Z
LED + QC	Exposed to intense LED light and administered with 20 mg/kg BW of quercetin
LED + QCG1	Exposed to intense LED light and administered with 10 mg/kg BW of quercetagetin
LED + QCG2	Exposed to intense LED light and administered with 20 mg/kg BW of quercetagetin
LED + L/Z + QCG (1:1)	Exposed to intense LED light and administered with 20 mg/kg BW of L/Z & Quercetagetin at a 1:1 ratio
LED + L/Z + QCG (1:0.5)	Exposed to intense LED light and administered with 20 mg/kg BW of L/Z & Quercetagetin at a 1:0.5 ratio
LED + L/Z + QCG (0.5:0.5)	Exposed to intense LED light and administered with 20 mg/kg BW of L/Z & Quercetagetin at a 0.5:0.5 ratio

Products were given by oral gavage for 28 days to rats before the intense LED light exposure for 48 h. LED: light-emitting diode light, Lutein/Zeaxanthin (Lutemax 2020 Free L/Z OS 20%/SF/IN-802); QC: quercetin; QCG: quercetagetin.

**Table 2 pharmaceuticals-16-01543-t002:** The content of total xanthophyll, lutein, zeaxanthin, and quercetagetin of products, % (*w/w*).

Product	Batch No	Total Xanthophyll	Lutein	Zeaxanthin	Quercetagetin
Lutemax 2020 (Free L/Z OS 20%/SF/IN-802)	0000090850	26.06	22.05	4.01	-
Quercetagetin	ND-93-111	-	-	-	80.92
Lutemax 2020 OS & Quercetagetin ratio 1:1	ND-93-139	28.51	22.93	5.14	26.69
Lutemax 2020 OS & Quercetagetin ratio 1:0.5	ND-93-141	28.86	23.27	5.16	12.52
Lutemax 2020 OS & Quercetagetin ratio 0.5:0.5	ND-93-140	16.34	13.14	2.88	11.56

## Data Availability

Data is contained within the article and Appendix A.

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
