# Peer review of "Lutein/Zeaxanthin Isomers and Quercetagetin Combination Safeguards the Retina from Photo-Oxidative Damage by Modulating Neuroplasticity Markers and the Nrf2 Pathway"

_pharmaceuticals, 2023, doi:10.3390/ph16111543_

Round 1

Reviewer 1 Report

The manuscript submitted by Sahin et al. has tried to evaluate the retinoprotective effects of two plant-derived substances, lutein and quercetagetin either alone or in combination in preventing photo-oxidative retinal damage, more specifically the loss of photoreceptor cell layer (ONL) and suggest their utility for therapeutic considerations. The protective effects were attributed to better photoreceptor survival via Nrf2 pathway-mediated reduction in oxidative stress and neuroplasticity modulation in the retina.

The concept of combining lutein and quercetagetin is rational, however, the provided results are not sufficient as convincing evidence to prove the hypothesis. This manuscript has major problems regrading design, data collection, presentation and interpretation. There is no visual function data collected, it would enhance the study if the data can be presented. There is no validation on retinal sections on these up or down regulated genes/proteins. There is no evidence to support authors’ claim that ONL was increased, which is not possible. 

The authors should explain why they started the therapy 28 days prior to the exposure to LED and provide citations if there are similar studies. Is this a protective or a rescue therapy? Why didn’t they continue the therapy during LED exposure?  Animals were euthanized 48 hours after subjected to LED, that is too short to see reduction of ONL. Longer time should be studied.

How can they evaluate the individual effects of lutein alone or in combination with where the product Lutemax 2020 is a mixture of XanthophyllLutein, and Zeaxanthin and % (w/w) of Xanthophyll > Lutein? Authors are requested to change the Lutein (L) treatment name as well so that the readers will not be confused that the protection is cumulative effects of XanthophyllLutein, and Zeaxanthin together instead of Lutein alone. 

Continuous exposure to intense LED light for 48 hours can cause severe stress and change the daily activity rhythm, activity, as well as the sleep/wake cycle, and may alter serum MDA levels, antioxidant enzymes (SOD, CAT, and GPx), and cytokine profile. Further, serum profiling of these parameters can be from any organ in addition to the retina. Hence, the authors are suggested to measure these parameters in retina tissue to confirm the retina as the source. In-gel activity assays can be performed for SOD, CAT, and GPx and genes of some cytokines and their receptors can be evaluated using qPCR. 

The authors are also requested to provide the individual values of statistical significance (p-values) for all the conditions (a-h) in bar graphs in the figure legend below each figure.

Both positive and negative error bars should be provided in all the bar graphs.

Figure 3: 

Bcl-2 and Bax are well-known anti-apoptotic and proapoptotic markers respectively. The authors are requested to explain why higher levels of Bcl-2 and lower levels of Bax and Caspase-3 were documented in the LED-exposed rats compared to the control. In the treated group LED+L+QCG (1:1) also lower levels of Bcl-2 and higher levels of Bax and Caspase-3 were documented. Are the authors concluding increased apoptosis in control and LED+L+QCG (1:1) groups compared to the LED-exposed group? 

Further, ab13585 detects both pro (full-length) and active (cleaved) Caspase 3. Pro Caspase 3 is detected at ~32 kDa. Active/cleaved Caspase 3 (large subunit) is detected at ~14-21 kDa as one or more bands. Hence, the authors are requested to show both the bands and provide the full blot images for the Caspase-3.

Figure 4.

Authors are requested to assess the activation (Keap1 and/or phosphorylation) and nuclear translocation/localization of NRF2 in the retinal tissue using immunofluorescence as this is the only mechanism (also the focus of the study) through which the treatment rescues photoreceptors (ONL) from photo-oxidative damage. 

The authors are also requested to check the p65-NFKB/total NFKB using WB to show the activation of NFKB and its nuclear localization using immunofluorescence in ONL.

Figure 6:

With the provided H/E histological images, the edema in the ganglion layer and the thickening of the outer plexiform layer are not clear in LED retina and the authors are requested to provide high quality histological images with proper labels and scale bars. However, disorganized INL with few cells, looks like a histologic artifact in the LED-exposed group! What parameters the authors considered and how did they measure the edema in the ganglion layer? The authors are also requested to provide brief details of ONL thickness measurement (how many sections/retina, how many retina/treatment).

Discussion:

Discussion is over-speculative in many places and some of the citations are not appropriate. The authors are requested to improve the discussion with the necessary results as supportive evidence. 

Author Response

The manuscript submitted by Sahin et al. has tried to evaluate the retinoprotective effects of two plant-derived substances, lutein and quercetagetin either alone or in combination in preventing photo-oxidative retinal damage, more specifically the loss of photoreceptor cell layer (ONL) and suggest their utility for therapeutic considerations. The protective effects were attributed to better photoreceptor survival via Nrf2 pathway-mediated reduction in oxidative stress and neuroplasticity modulation in the retina.

The concept of combining lutein and quercetagetin is rational, however, the provided results are not sufficient as convincing evidence to prove the hypothesis. This manuscript has major problems regrading design, data collection, presentation and interpretation. There is no visual function data collected, it would enhance the study if the data can be presented. There is no validation on retinal sections on these up or down regulated genes/proteins. There is no evidence to support authors’ claim that ONL was increased, which is not possible.

Response: Thanks to the reviewer for him/her thorough review of our manuscript. We recognize the importance of validating these up or down-regulated genes/proteins in retinal sections to ensure the robustness of our conclusions. However, we have shown histopathological and Western blot results that are reliable and acceptable for preclinical experimental studies. Please check Figure 6 (especially, outer nuclear layer (ONL) thickness bars graph)  again in case you missed it. In accordance that the reviewer is right; the ONL increse statement was written inadvertently and is corrected as follows: " prevented the reduction of ONL thickness in LED-induced retinal damage rats." in the abstract section. In addition, as we claim that the reduction of ONL thickness following LED exposure (proved by many studies; Orhan, Gencoglu et al., 2021; Koyama et al., 2019) was prevented by phytochemicals. Also, there are studies that quercetin (similar to quercetagetin) (Koyama et al., 2019) and lutein (Yang et al., 2020) can prevent the decrease of ONL thickness. We revised the result section as “prevented the decrease of ONL thickness” in section 2.4.

The authors should explain why they started the therapy 28 days prior to the exposure to LED and provide citations if there are similar studies. Is this a protective or a rescue therapy? Why didn’t they continue the therapy during LED exposure? Animals were euthanized 48 hours after subjected to LED, that is too short to see reduction of ONL. Longer time should be studied.

Response: Thanks to the reviewer. As stated in the introduction section, we aimed to investigate the protective effect of lutein and quercetagetin (Similar studies were cited in section 4.2). Therefore, we did not continue the therapy during led exposure to obtain more objective results. We used a similar method in our previous study (Orhan, Gencoglu et al., 2021). Also, many experimental studies that investigated the protective effects of different compounds on the retina have similar designs as to the present study (Yang et al., 2020, Amato et al., 2021). As reported in our previous studies, 48 hours of LED exposure is enough to reduce ONL (Orhan, Gencoglu et al., 2021; Orhan, Tuzcu et al., 2021). Additionally, Koyomo et al. (2019) reported that even after one day following 24 h light exposure, ONL thickness can decrease. Section 4.2. revised to eliminate confusion. Krigel et al. (2016) reported that  after 24hrs of continuous exposure of rats with dilated pupils, to white-cold LED at 500 lux (we used 750 lux), a significant reduction of ONL thickness was found not only in albinos but also, to a lesser extent in pigmented rats (https://doi.org/10.1016/j.neuroscience.2016.10.015).

How can they evaluate the individual effects of lutein alone or in combination with where the product Lutemax 2020 is a mixture of Xanthophyll, Lutein, and Zeaxanthin and % (w/w) of Xanthophyll > Lutein? Authors are requested to change the Lutein (L) treatment name as well so that the readers will not be confused that the protection is cumulative effects of Xanthophyll, Lutein, and Zeaxanthin together instead of Lutein alone.

Response: Thanks to the reviewer. It was corrected as lutein and zeaxanthin (L/Z, Lutemax 2020). As it is known lutein and zeaxanthin are important xanthophylls. In Table 2 xanthophyll means the total xanthophyll (Lutein, and Zeaxanthin) for Lutemax. Table 2 revised.

Continuous exposure to intense LED light for 48 hours can cause severe stress and change the daily activity rhythm, activity, as well as the sleep/wake cycle, and may alter serum MDA levels, antioxidant enzymes (SOD, CAT, and GPx), and cytokine profile. Further, serum profiling of these parameters can be from any organ in addition to the retina. Hence, the authors are suggested to measure these parameters in retina tissue to confirm the retina as the source. In-gel activity assays can be performed for SOD, CAT, and GPx and genes of some cytokines and their receptors can be evaluated using qPCR.

Response: Thank you for the suggestion. We prioritized targeted analyses due to the limited amount of retinal tissue available from each animal. Therefore, we focused on the most common parameters, such as Nrf2 and HO-1 for the determination of retinal antioxidant status at the molecular levels.

The authors are also requested to provide the individual values of statistical significance (p-values) for all the conditions (a-h) in bar graphs in the figure legend below each figure.

Response: The figure legends were revised and clarified in detail

Both positive and negative error bars should be provided in all the bar graphs.

Response: The figures were revised.

Figure 3:

Bcl-2 and Bax are well-known anti-apoptotic and proapoptotic markers respectively. The authors are requested to explain why higher levels of Bcl-2 and lower levels of Bax and Caspase-3 were documented in the LED-exposed rats compared to the control. In the treated group LED+L+QCG (1:1) also lower levels of Bcl-2 and higher levels of Bax and Caspase-3 were documented. Are the authors concluding increased apoptosis in control and LED+L+QCG (1:1) groups compared to the LED-exposed group?

Response: Thanks to the reviewer. Unexpected regulation of Bcl-2 and Bax may not always result in tissue damage because each molecular pathway's effect is unclear, nor is there conclusive evidence that Bax stimulates apoptosis and Bcl-2 prevents apoptosis. In addition, apoptosis does not always cause tissue damage. The possible reasons are added in the discussion section (second paragraph) .

Further, ab13585 detects both pro (full-length) and active (cleaved) Caspase 3. Pro Caspase 3 is detected at ~32 kDa. Active/cleaved Caspase 3 (large subunit) is detected at ~14-21 kDa as one or more bands. Hence, the authors are requested to show both the bands and provide the full blot images for the Caspase-3.

Response: The reviewer is right. In this study, we considered the active caspase form determined at the level of approximately 17 kDa and evaluated it in the manuscript. The full immunoblot can be found in the supplementary file data. Please check the Supplementary Fig. 1 (Panel C).

Figure 4.

Authors are requested to assess the activation (Keap1 and/or phosphorylation) and nuclear translocation/localization of NRF2 in the retinal tissue using immunofluorescence as this is the only mechanism (also the focus of the study) through which the treatment rescues photoreceptors (ONL) from photo-oxidative damage.

The authors are also requested to check the p65-NFKB/total NFKB using WB to show the activation of NFKB and its nuclear localization using immunofluorescence in ONL.

Response: Thank you for the recommendation. The use of the suggested immunofluorescence techniques will be taken into consideration in our future studies.

Figure 6:

With the provided H/E histological images, the edema in the ganglion layer and the thickening of the outer plexiform layer are not clear in LED retina and the authors are requested to provide high quality histological images with proper labels and scale bars. However, disorganized INL with few cells, looks like a histologic artifact in the LED-exposed group! What parameters the authors considered and how did they measure the edema in the ganglion layer? The authors are also requested to provide brief details of ONL thickness measurement (how many sections/retina, how many retina/treatment).

Response: Figure 6 revised as suggested. Retinas from seven rats of each group were taken for measurements of the ONL and mean ONL values were determined by an independent blinded pathologist. It was detailed in the 4.5 Histopathological Analysis section.

Discussion:

Discussion is over-speculative in many places and some of the citations are not appropriate. The authors are requested to improve the discussion with the necessary results as supportive evidence.

Response: The discussion section was revised as suggested.

Reviewer 2 Report

The manuscript submitted for review presents the results of the influence of nutritional supplements on the state of the retina. In the work, the authors used a model of rats with light-induced LED retinopathy. They tested the impact of lutein (L) and quercetagetin (QCG) alone or in combination over a period of 28 daysRats were exposed to LED light for 48 hours. n=7 in group was used.  In rats, after LED  exposure, the level of cytokines and MDA sharply increased, both systemically and in the retina. It also reduced serum antioxidant enzyme activities and retinal Nrf2, HO-1, GAP43, NCAM, and outer nuclear layer (ONL) thickness. In groups that received nutritional supplements before exposure, the negative effects of light were reduced. A combined L and QCG diet exhibited  most protective  effect on the retina.

The authors rightly point out that defense mechanisms are mediated by the modulation of neuroplasticity markers and nuclear transcription factors in the rats' retinal cells such as Nrf2.

I agree with the conclusions of the authors.

As a small note:

I think that in the abstract it is worth indicating the working dosages.

Author Response

The manuscript submitted for review presents the results of the influence of nutritional supplements on the state of the retina. In the work, the authors used a model of rats with light-induced LED retinopathy. They tested the impact of lutein (L) and quercetagetin (QCG) alone or in combination over a period of 28 days. Rats were exposed to LED light for 48 hours. n=7 in group was used. In rats, after LED  exposure, the level of cytokines and MDA sharply increased, both systemically and in the retina. It also reduced serum antioxidant enzyme activities and retinal Nrf2, HO-1, GAP43, NCAM, and outer nuclear layer (ONL) thickness. In groups that received nutritional supplements before exposure, the negative effects of light were reduced. A combined L and QCG diet exhibited  most protective  effect on the retina.

The authors rightly point out that defense mechanisms are mediated by the modulation of neuroplasticity markers and nuclear transcription factors in the rats' retinal cells such as Nrf2.

I agree with the conclusions of the authors.

As a small note:

I think that in the abstract it is worth indicating the working dosages.

Response: Thanks to the reviewer. The dosages were added in the abstract.

Reviewer 3 Report

1.       For histograms in figures, please display all individual points and add n number in the figure legend part. Label all group information and statistic results in each panel. The capitals (a-h) above the bars are confusing.

2.       Line 330, from published paper entitled “Duration of Mydriasis Produced by 0.5% and 1% Tropicamide in Sprague–Dawley Rats” says “The duration of action is at least 5 h for 0.5% tropicamide and 6 h for 1% tropicamide.” Were the 1% tropicamide applied to rats every 6 hours during light exposure or it’s only a one-time application?

3.       Line 124, LED exposed rats have lower level of Caspase-3 compared to controls. Caspase-3 is a marker for apoptosis, this means LED exposed rats have less apoptosis cells compared to healthy controls, which is contradict with pushed literatures. The authors should discuss on this discrepancy.

4.       Please add scale to the images and label each image. Please indicate the location of the retinal area for comparison and highlight the edema regions.

5.       For statistics part, please consider use Bonferroni Correction for multiple comparisons.

Author Response

  1. For histograms in figures, please display all individual points and add n number in the figure legend part. Label all group information and statistic results in each panel. The capitals (a-h) above the bars are confusing.

Response: Thanks to the reviewer. Figures and legends were revised as requested.

  1. Line 330, from published paper entitled “Duration of Mydriasis Produced by 0.5% and 1% Tropicamide in Sprague–Dawley Rats” says “The duration of action is at least 5 h for 0.5% tropicamide and 6 h for 1% tropicamide.” Were the 1% tropicamide applied to rats every 6 hours during light exposure or it’s only a one-time application?

Response: For establishing an LED-induced retinal damage model, one-time 1% tropicamide administration is commonly used and considered sufficient. Similar to Wang et al. (2017); Yang et al. (2020); Orhan, Tuzcu et al. (2021); Amato et al. (2021); Orhan, Gencoglu et al. (2021), we applied 1% tropicamide, for once.

  1. Line 124, LED exposed rats have lower level of Caspase-3 compared to controls. Caspase-3 is a marker for apoptosis, this means LED exposed rats have less apoptosis cells compared to healthy controls, which is contradict with pushed literatures. The authors should discuss on this discrepancy.

Response: The discussion section was revised.

  1. Please add scale to the images and label each image. Please indicate the location of the retinal area for comparison and highlight the edema regions.

Response:Figure 6 has been revised as suggested

  1. For the statistics part, please consider use Bonferroni Correction for multiple comparisons.

Response: The statistic part was revised as requested. Changing the statistical method did not lead to substantial alterations in results. Figures and legends were revised.

Reviewer 4 Report

Sahin and coauthors presented a study on the combination of lutein and quercetagetin safeguards the retina from photo-oxidative damage by modulating neuroplasticity markers and the Nrf2 pathway. Their results demonstrated that a 1:1 mixture of lutein and quercetagetin exhibited the greatest protective effects on the retina.

Overall, the manuscript is not easy to follow when compared to the simple results highlighted in the study. The conclusion is simplistic, and the research only includes serum analyses, Western blotting, and histological measurements. The study appears overly complicated, with too many groups that seem unnecessary.

The design of their study could be improved for better comprehension, unless there is a very specific reason for showing all the data simultaneously. For instance, they could introduce an LED+GCG2 group, consistent with other treatment doses (20 mg/kg), and include supplementary data for the comparison between LED+QCG1 and LED+QCG2. The LED+L+QCG (1:1) group has shown promising results, and this group could be emphasized within the main findings of the study. Additionally, the authors might think about providing separate supplementary data that compares different concentrations, such as 1:1, 1:0.5, and 0.5:0.5. By doing so, they could streamline their results, concentrating on the positive control, negative control, and four specific treatment groups (L, QC, QCG, and L+QCG). Even more, it unclear whether the inclusion of the LED+QC group is warranted, especially since the only comparison being made involves the L+QCG mixture.

The experiments conducted by the authors, including serum analyses, Western blotting, and histological measurements, are not enough to substantiate their hypothesis. Furthermore, the description provided in the materials and methods section is inadequate, particularly regarding sample collection and histopathological evaluation. For instance, while the authors measured the outer nuclear layer thickness as part of their data analysis using histopathology, they failed to provide details about the measurement location, the number of eyes used for the analysis, and the methods employed. Since retinal thickness can vary depending on the location within the eye, it is essential that the authors clarify every single detail related to these measurements.

Both Western blotting and histological evaluation are semiquantitative methods. I suggest that the authors narrow the focus to a few principal groups (such as L, QCG, and L+QCG) and enhance their findings with additional experimental data. This could include techniques like mRNA expression or immunohistochemistry to lend more robust support to their hypothesis.

I'm uncertain about the benefit of using abbreviations for single words, such as "L" for lutein and "Q" for quercetin. Utilizing "L" for lutein in the introduction and discussion sections makes the manuscript more challenging to read. Additionally, the use of abbreviations is not consistent throughout the document.

More detailed information needs to be added to all figure legends. Additionally, in the statistical analysis, some bars are marked with 2-3 alphabets simultaneously. I cannot understand what this signifies, and it needs to be explained in the figure legend.

Line 3: please remove “g”

Line 35: the results of this study are not enough to support the regenerative effects on the retina.  

Line 95: please change “GPx” to “GPX”

Line 205: please italicize “et al.”

Line 317: please italicize “Tagetes erecta”

Line 317: please add a reference for the sentence “No chronic toxicity was detected in animals or humans, and no mutagenicity was found in any of the products”

Line 321-322: please check for missing or doubling periods at the end of the sentences.

Figure 6: please add scale bar for all images.

The overall English in the manuscript is good. However, some of the very long sentences could be improved by separating them into two distinct sentences for clarity.

Author Response

Sahin and coauthors presented a study on the combination of lutein and quercetagetin safeguards the retina from photo-oxidative damage by modulating neuroplasticity markers and the Nrf2 pathway. Their results demonstrated that a 1:1 mixture of lutein and quercetagetin exhibited the greatest protective effects on the retina.

Overall, the manuscript is not easy to follow when compared to the simple results highlighted in the study. The conclusion is simplistic, and the research only includes serum analyses, Western blotting, and histological measurements. The study appears overly complicated, with too many groups that seem unnecessary.

Response: We do not agree with the Reviewer. There are thousands of similar studies conducted to reveal the effects of retinal damage. The research will, of course, include serum analyses, Western blot and histological measurements. I couldn't quite understand what else it would contain. Because these methods provide the basis for investigating the mechanisms we focused on. We understand that simplicity is often key in scientific communication, and we aim to strike a better balance between presenting comprehensive results and making them easy to follow.

The design of their study could be improved for better comprehension, unless there is a very specific reason for showing all the data simultaneously. For instance, they could introduce an LED+GCG2 group, consistent with other treatment doses (20 mg/kg), and include supplementary data for the comparison between LED+QCG1 and LED+QCG2. The LED+L+QCG (1:1) group has shown promising results, and this group could be emphasized within the main findings of the study. Additionally, the authors might think about providing separate supplementary data that compares different concentrations, such as 1:1, 1:0.5, and 0.5:0.5. By doing so, they could streamline their results, concentrating on the positive control, negative control, and four specific treatment groups (L, QC, QCG, and L+QCG). Even more, it unclear whether the inclusion of the LED+QC group is warranted, especially since the only comparison being made involves the L+QCG mixture.

Response: Thanks to the reviewer. When the reviewer mentioned the "LED+GCG2" group, him/her likely meant the "low dose LED+QC" group. We intended to emphasize the distinction between QC and QCG. Thus, we exclusively proceeded with the LED+QC group, excluding the low dose. Therefore, introducing a low QC group would be unnecessary for the specific objectives of our study.

The LED+L+QCG (1:1) group was emphasized within the results section, as suggested.

The result section was extended, and all statistical comparisons were indicated in the legend of the figures. Additionally, the requested comparisons were done between the   LED+L+QCG (1:1), LED+L+QCG (1:0.5), and LED+L+QCG (0.5:0.5) and presented in the supplementary file.

The experiments conducted by the authors, including serum analyses, Western blotting, and histological measurements, are not enough to substantiate their hypothesis. Furthermore, the description provided in the materials and methods section is inadequate, particularly regarding sample collection and histopathological evaluation. For instance, while the authors measured the outer nuclear layer thickness as part of their data analysis using histopathology, they failed to provide details about the measurement location, the number of eyes used for the analysis, and the methods employed. Since retinal thickness can vary depending on the location within the eye, it is essential that the authors clarify every single detail related to these measurements.

Response: The descriptions provided in the materials and methods section, particularly regarding sample collection and histopathological evaluation, were detailed as suggested.

Both Western blotting and histological evaluation are semiquantitative methods. I suggest that the authors narrow the focus to a few principal groups (such as L, QCG, and L+QCG) and enhance their findings with additional experimental data. This could include techniques like mRNA expression or immunohistochemistry to lend more robust support to their hypothesis.

Response: We have some concerns about narrowing the focus of this study. We would prefer not to restrict the groups, as doing so may lead us away from the primary aim and objectives of the study because our main purpose is demonstrating different dosage formulations of L and QCG.

Although western blotting is a semiquantitative method, it offers unique advantages in studying protein-related aspects of biology that mRNA expression cannot fully address. Western blotting's ability to provide direct insights into protein expression, modifications, and interactions makes it an essential tool for experimental research. Western blotting allows direct visualization and analysis of protein levels, post-translational modifications, and protein interactions. This is crucial for understanding the functional aspects of proteins, which cannot be fully captured by studying gene expression alone. Western blotting is the final analysis to measure protein levels for confirming gene expression findings obtained through techniques like real-time PCR. Therefore, we only used the western blotting method to represent our results more accurately. We will consider the reviewer's advanced suggestions in the future in detail.

I'm uncertain about the benefit of using abbreviations for single words, such as "L" for lutein and "Q" for quercetin. Utilizing "L" for lutein in the introduction and discussion sections makes the manuscript more challenging to read. Additionally, the use of abbreviations is not consistent throughout the document.

Response: All abbreviations were revised for enhanced readability, as recommended. But we used QC for quercetin; it is an appropriate abbreviation.

More detailed information needs to be added to all figure legends. Additionally, in the statistical analysis, some bars are marked with 2-3 alphabets simultaneously. I cannot understand what this signifies, and it needs to be explained in the figure legend.

Response: Figure legends were revised.

Line 3: please remove “g”

Response: Removed.

Line 35: the results of this study are not enough to support the regenerative effects on the retina.  

Response: The sentence was revised.

Line 95: please change “GPx” to “GPX”

Response: Revised.

Line 205: please italicize “et al.”

Response: Revised.

Line 317: please italicize “Tagetes erecta”

Response: Revised.

Line 317: please add a reference for the sentence “No chronic toxicity was detected in animals or humans, and no mutagenicity was found in any of the products”

Response: The sentence was revised and the references were added.

Line 321-322: please check for missing or doubling periods at the end of the sentences.

Response: Checked.

Figure 6: please add scale bar for all images.

Response:Added.

Comments on the Quality of English Language

The overall English in the manuscript is good. However, some of the very long sentences could be improved by separating them into two distinct sentences for clarity.

Response: Thanks to the reviewer. Revised as suggested.

Reviewer 5 Report

In this paper, the authors demonstrated the effect of plant-derived substances, lutein and quecetagetin on LED-induced retinal damage by using rat model. They showed that lutein and quecetagetin treatment reduced LED-induced retinal damage. They also showed LED-induced up-regulation of serum malondialdehyde and inflammatory cytokine levels, and reduced anti-oxidant enzyme levels were also ameliorated by  lutein and quecetagetin treatment. These results indicate that lutein and quecetagetin are promising drugs for light-induced retinal damage treatment.

Although the almost data are reasonable and the manuscript is well written, the following points should be clarified before acceptance in Pharmaceuticals.

1) In LED exposed retinal tissues down-regulated Bax and caspase-3, and induced Bcl2 and lutein and quercetagetin attenuate these effects (Fig. 3). These results indicate that LED light exposure decreases apoptosis, and lutein and quercetagetin diminish anti-apoptotic effect of LED light. 

On the contrary to this study, other light-induced retinal damage model rather induced retinal apoptosis (Ref. 19, 20). The authors should discuss the difference between other light-induced retinal damage models in detail.

2) Figure 3C, Upon activation, caspase-3 is cleaved for its activation and induces apoptosis. Is caspase-3 band showed here is full length caspase-3 or cleaved caspase-3?

3) Figure 4C, according to the information described in Materials and Methods, NF-kB antibody used here specifically recognizes NF-kB p65 subunit acetylated at lysine310. It should be described correctly.

Author Response

In this paper, the authors demonstrated the effect of plant-derived substances, lutein and quecetagetin on LED-induced retinal damage by using rat model. They showed that lutein and quecetagetin treatment reduced LED-induced retinal damage. They also showed LED-induced up-regulation of serum malondialdehyde and inflammatory cytokine levels, and reduced anti-oxidant enzyme levels were also ameliorated by  lutein and quecetagetin treatment. These results indicate that lutein and quecetagetin are promising drugs for light-induced retinal damage treatment.

Although the almost data are reasonable and the manuscript is well written, the following points should be clarified before acceptance in Pharmaceuticals.

Response: Thanks to the reviewer for him/her thorough review of our manuscript.

1) In LED exposed retinal tissues down-regulated Bax and caspase-3, and induced Bcl2 and lutein and quercetagetin attenuate these effects (Fig. 3). These results indicate that LED light exposure decreases apoptosis, and lutein and quercetagetin diminish anti-apoptotic effect of LED light. 

On the contrary to this study, other light-induced retinal damage model rather induced retinal apoptosis (Ref. 19, 20). The authors should discuss the difference between other light-induced retinal damage models in detail.

Response: Thanks to the reviewer. The possible reason for the unexpected regulation of Bcl-2/Bax/Caspase-3 interaction is discussed in the discussion section.

2) Figure 3C, Upon activation, caspase-3 is cleaved for its activation and induces apoptosis. Is caspase-3 band showed here is full length caspase-3 or cleaved caspase-3?

Response: Thanks to the reviewer. We considered the active/cleaved caspase form determined in the manuscript. The full immunoblot can be found in the supplementary file data. Please find the Suppplementary Fig. 1 (Panel C).

3) Figure 4C, according to the information described in Materials and Methods, NF-kB antibody used here specifically recognizes NF-kB p65 subunit acetylated at lysine310. It should be described correctly.

Response: Thanks to the reviewer. NF-kB p65 is described correctly in the material and methods section and Figure 4C.

Round 2

Reviewer 1 Report

Authors have  addressed my comments. Please see the attaced file for minor correction

Author Response

Reviewer 1

The authors have addressed my comments.

Minor corrections

Figure6:A and I should be replaced with better laminated images. All images should be taken at the same power,only one scale bar is needed. Label retinal layers.

Response: Thanks to the reviewer. Revised as suggested

Reviewer 3 Report

The authors have significantly improved the manuscript. However, the statistic results in figure legends are not straightforward to read, please sonsider listing P values of multiple comparisons in tables. 

Author Response

Reviewer 3

The authors have significantly improved the manuscript. However, the statistic results in figure legends are not straightforward to read, please sonsider listing P values of multiple comparisons in tables.

Response: Thanks to the reviewer. The statistical comparisons (Table S1-6) presented in supplementary material.

Reviewer 4 Report

Sahin and coauthors addressed most of the previous comments appropriately, and the manuscript looks improved.

Author Response

Reviewer 4

Sahin and coauthors addressed most of the previous comments appropriately, and the manuscript looks improved.

Response: Thanks to the reviewer.

Reviewer 5 Report

The reviewer appreciates for conscientious response to the comments.

The reviewer thinks the manuscript is properly revised, so the manuscript would be acceptable for Pharmaceuticals.

Author Response

Reviewer 5

The reviewer appreciates for conscientious response to the comments. The reviewer thinks the manuscript is properly revised, so the manuscript would be acceptable for Pharmaceuticals. 

Response: Thanks to the reviewer.